# Three-Dimensional Microfibrous Scaffold with Aligned Topography Produced via a Combination of Melt-Extrusion Additive Manufacturing and Porogen Leaching for In Vitro Skeletal Muscle Modeling

**DOI:** 10.3390/bioengineering11040332

**Published:** 2024-03-28

**Authors:** Mattia Spedicati, Alice Zoso, Leonardo Mortati, Valeria Chiono, Elena Marcello, Irene Carmagnola

**Affiliations:** 1Department of Mechanical and Aerospace Engineering, Politecnico di Torino, 10129 Torino, Italy; mattia.spedicati@polito.it (M.S.); alice.zoso@polito.it (A.Z.); valeria.chiono@polito.it (V.C.); 2POLITO BioMedLab, Politecnico di Torino, 10129 Torino, Italy; 3Interuniversity Center for the Promotion of the 3Rs Principles in Teaching and Research, 56122 Pisa, Italy; 4Istituto Nazionale di Ricerca Metrologica (INRIM), 10135 Torino, Italy; l.mortati@inrim.it

**Keywords:** Melt-Extrusion Additive Manufacturing, PCL, PEG, porogen leaching, fibrous scaffold, skeletal muscle tissue, in vitro modeling

## Abstract

Skeletal muscle tissue (SMT) has a highly hierarchical and anisotropic morphology, featuring aligned and parallel structures at multiple levels. Various factors, including trauma and disease conditions, can compromise the functionality of skeletal muscle. The in vitro modeling of SMT represents a useful tool for testing novel drugs and therapies. The successful replication of SMT native morphology demands scaffolds with an aligned anisotropic 3D architecture. In this work, a 3D PCL fibrous scaffold with aligned morphology was developed through the synergistic combination of Melt-Extrusion Additive Manufacturing (MEAM) and porogen leaching, utilizing PCL as the bulk material and PEG as the porogen. PCL/PEG blends with different polymer ratios (60/40, 50/50, 40/60) were produced and characterized through a DSC analysis. The MEAM process parameters and porogen leaching in bi-distilled water allowed for the development of a micrometric anisotropic fibrous structure with fiber diameters ranging from 10 to 100 µm, depending on PCL/PEG blend ratios. The fibrous scaffolds were coated with Gelatin type A to achieve a biomimetic coating for an in vitro cell culture and mechanically characterized via AFM. The 40/60 PCL/PEG scaffolds yielded the most homogeneous and smallest fibers and the greatest physiological stiffness. In vitro cell culture studies were performed by seeding C2C12 cells onto a selected scaffold, enabling their attachment, alignment, and myotube formation along the PCL fibers during a 14-day culture period. The resultant anisotropic scaffold morphology promoted SMT-like cell conformation, establishing a versatile platform for developing in vitro models of tissues with anisotropic morphology.

## 1. Introduction

Skeletal muscle tissue (SMT) represents 40–45% of the adult human body [1,2], with its main function being to generate contractile forces to control body movements and support internal organs [3]. These capabilities are possible thanks to the hierarchical and anisotropically oriented structure of SMT [4,5]. A single muscle is composed of multiple fascicles with diameters ranging between 0.5 and 30 mm [5]. Each fascicle is made up of many muscle fibers (within 20 to 100 μm in diameter) [6], together with neurons and blood vessels, forming a cylindrical bundle. A muscle fiber, or myofiber, is a multinucleated muscle cell derived from the fusion of myoblasts during myogenesis [3,4] and is covered by the endomysium, mostly composed of laminin and collagen type IV [6,7,8]. Each myofiber contains many myofibrils (1–2 μm diameter) [5], thread-like structures organized in repeated sections of sarcomeres that are the basic muscle cell units and contain specific proteins, actin and myosin, interspaced with Z disks (also called z-lines) [9].

Skeletal muscle functionality can be severely impaired by a range of factors caused by disease conditions or traumatic injuries [2,10]. The demand for efficient strategies for the restoration of impaired muscle function and the exploration of novel therapeutic avenues has increasingly stimulated interest in fields such as tissue engineering and regenerative medicine [2,3]. In this context, Skeletal Muscle Tissue Engineering (SMTE) can provide tools for the in vitro modeling of healthy and pathological SMT, which is useful for discovering and testing new drugs and therapies [11].

To replicate the native morphology of SMT hierarchical structure, scaffolds must possess a precisely aligned anisotropic tridimensional architecture capable of providing cues for myoblast alignment [4,5,6,8]. This alignment is essential for the formation of high-density cell sheets, which can subsequently undergo fusion and differentiate into myotubes, ultimately resulting in the development of functional myofibers [12].

The optimization of myoblast organization to enhance cell–cell interactions and the maximization of the cell–substrate contact area are critical factors in promoting myogenesis [8]. The literature reports that surface stiffness influences myotube formation, with an optimal substrate stiffness value of approximately 12 kPa [13]. Indeed, to reproduce substrate aligned topography, surface composition and stiffness are crucial for reproducing native stimuli and for supporting cell adhesion and proliferation. Typically, the SMT extracellular matrix (ECM) exhibits a high content of collagen [14] and a stiffness, measured via Atomic Force Microscopy (AFM), ranging between 10 and 60 kPa [15,16].

Studies have demonstrated that the architecture of aligned nanofibrillar scaffolds promotes cytoskeletal organization along the alignment direction, aiding cell alignment and elongation compared to randomly oriented scaffolds [17,18]. To address cell morphology and organization, several scaffold manufacturing strategies can be exploited, such as microscale topography achieved through micropatterned substrates [8,19,20], aligned nanofibrous 2D matrices that mimic the morphology of native ECM proteins [8,21], and 3D scaffolds featuring an anisotropic architecture and porosity, favoring myoblasts organization into wide and elongated myotubes [8]. Micropatterned substrates are characterized by microscale grooves, ridges, posts, and holes within which myoblasts can organize, fuse, and differentiate into myotubes and then myofibers [9,22]. The production methods commonly used for micropatterned substrates are photolithography and soft lithography [8,12,20]. Nanoscale 2D fibrous scaffolds are commonly used due to their ability to mimic the aspect ratio and size scale of ECM proteins [23], such as collagen fibers, whose diameters range between 260 and 410 nm [22]; moreover, their high surface-area-to-volume ratio promotes cell adhesion, migration, proliferation, and differentiation [8]. Aligned nanofibers, mostly fabricated using electrospinning [8,21], encourage myoblasts alignment, promoting myogenesis and representing a critical first step toward successful muscle tissue engineering [21,23,24]. However, 2D aligned scaffolds fall short in reproducing the 3D structure of muscle fascicles (0.5–30 mm in diameter) [5], providing bi-dimensional substrates, thus limiting their ability to replicate the complex 3D hierarchy of the SMT [5,6]. Tridimensional anisotropically aligned porous scaffolds have been conventionally produced using Thermally Induced Phase Separation (TIPS) followed by lyophilization [25] or freeze-drying, applying a unidirectional freezing gradient [26]. However, the lack of reproducibility associated with these conventional techniques represents the main limitation of these methodologies. On the contrary, additive manufacturing techniques, including Melt Extrusion Additive Manufacturing (MEAM), have garnered attention due to their versatility in fabricating complex microscale substrates with high reproducibility [27,28,29]. Recently, a combination of MEAM with porogen leaching has been investigated to fabricate three-dimensional polymeric scaffolds with a fibrous structure and anisotropic morphology [30,31]. Contrarily to the conventional porogen leaching process used to produce non-oriented pores [32,33,34], the application of shear stresses during the extrusion process can be employed to achieve an aligned porogen phase within the bulk polymer, depending on the porogen volume fraction. After the extrusion process, the immersion of the sample in a suitable solution allows the dissolution of the porogen phase, generating aligned porosities or aligned microfibers [30,31,35]. Among the broad range of materials exploited for MEAM, poly(ε-caprolactone) (PCL) is a thermoplastic polymer widely used in biological applications, as it is biodegradable and biocompatible [31,36]. Although PCL may lack high bioactivity due to its hydrophobicity, it shows remarkable versatility in terms of process techniques as well as the possibility of being functionalized through many approaches, allowing for the deposition of bioactive coatings [37]. PCL was studied as a matrix bulk material to create a porous structure via conventional processing techniques using a range of porogens [38,39,40,41], including water-soluble thermoplastic polymers such as polyethylene glycol (PEG) and polyethylene oxide (PEO) [42]. Limited work has been conducted on the combination of MEAM with porogen leaching using PCL as a bulk material and water-soluble thermoplastic polymers as porogens. Specifically, Hwangbo et al. blended PCL with polyvinyl alcohol (PVA) to produce fibrous structures that, combined with collagen and cellularized with murine-derived myoblast cells, were intended to reproduce muscle bundles [31,35]. The blends were processed to form parallel struts that, after porogen leaching, led to the formation of PCL bundle structures. Such structures were coated with collagen [35] or used as a mold for casting collagen hydrogel [31], and the final constructs were then cellularized with murine-derived myoblast cells. The aligned hierarchical structure and the presence of collagen induced high degrees of myoblast alignment and efficient myogenic differentiation. Even though the fibrous structures were able to reproduce the anisotropic hierarchy of SMT and sustain myotube formation, the manufacturing technique to achieve such structures was characterized by a complex multistep process [30,31,35]. Moreover, biomimicry was achieved using collagen, which, although highly biomimetic, involves high costs and immunological issues [43,44].

In this work, PCL 3D fibrous scaffolds with an aligned morphology were produced by combining MEAM and porogen leaching techniques, using PEG as a porogen agent. Homogeneous PCL/PEG blends were produced with different volume ratios using a solvent casting technique [41,45] and extruded under pneumatic pressure, promoting the orientation of polymeric phases. After porogen leaching, PCL microfibers showed diameters within the range of 10–60 µm, comparable to muscle fibers’ dimensions [6], and were homogeneously aligned with the printing direction. PCL fibrous scaffolds were biomimetically coated with Gelatin type A [43,46], a collagen derivative, one of the main components of the skeletal muscle ECM in both physiological and pathological conditions [14,47]. Afterwards, C2C12 cells were seeded onto PCL fibrous Gelatin-coated scaffolds and cultured for 14 days, demonstrating the scaffolds potential to induce cell attachment, orientation, and myotube formation. The obtained results demonstrated that the proposed PCL fibrous scaffolds are a promising tool for the development of SMTE in vitro models, as they provide a 3D biomimetic anisotropic support with fibers dimensionally comparable to muscle bundles.

## 2. Materials and Methods

### 2.1. Materials

Polycaprolactone (PCL), Mw = 43,000 Da, was purchased from Polysciences Europe GmbH, (Hirschberg an der Bergstrasse, Germany), while Polyethylene glycol (PEG), Mw = 35,000 Da, was purchased from Merk Life Science Srl (Milano, Italy). Chloroform, which was used as solvent, was purchased from Carlo Erba Reagents (Cornaredo, MI, Italy). Gelatin (G) type A, purchased from Merk Life Science Srl, was used as a coating for the cell culture of C2C12 murine cells purchased from ATCC (Manassas, VA, USA).

### 2.2. Polymeric Blend Production

PCL/PEG blend films with different polymer weight ratios (% *w*/*w*, PCL/PEG = 60/40, 50/50, 40/60) were produced by solvent casting. First, 20% *w*/*v* of PCL/PEG chloroform solutions were mixed for 3 h at 400 rpm and room temperature. Solutions were then cast into glass Petri dishes (6 cm in diameter) and dried for 4 days under a fume hood with low ventilation.

### 2.3. Polymeric Blend Characterization

Thermal properties of the starting materials (neat PCL and PEG) and blends (PCL/PEG 60/40, 50/50, and 40/60 films) were evaluated via Differential Scanning Calorimetric (DSC) analysis using an NEXTA^®^ DSC200 differential scanning calorimeter (Hitachi High-Tech Corporation, Tokyo, Japan). Approximately 5 mg of each sample was heated from −90 °C to 150 °C at a heating rate of 10 °C/min (heating cycle). NEXTA^®^ 2.7 Standard Analysis software was used to analyze the DSC thermographs, evaluating the melting temperature (T_m_) and the enthalpy of fusion (ΔH_m_).

### 2.4. Scaffold Design and Fabrication

Three-dimensional PCL/PEG scaffolds were produced by Melt Extrusion Additive Manufacturing (MEAM) using “INVIVO” 3D printer (Rokit Healthcare, Seoul, Republic of Korea) equipped with a 400 µm nozzle. A CAD model was designed with Autodesk “Inventor 2023^®^“ software (San Francisco, CA, USA) and exported in STL format. Printing parameters (fill density (FD) and print speed (PS)) were set directly in NewCreatorK slicing software (version 1.57.70, copyright 2017 by Rokit Healthcare).

PCL/PEG blend films were cut into small square pieces (average length of 4 mm) and used as starting materials for the printing process. Scaffolds were printed at 100 °C, 10 kPa pressure, and a PS of 2 mm/s. Scaffolds with two FDs (27% and 33%) were investigated, corresponding, respectively, to a larger (550 µm) and smaller (500 µm) spacing between parallel filaments (as shown in Figure 1a). The printed scaffolds (Figure 1a) measured 6.1 × 30 mm for an FD of 27% and 6.7 × 30 mm for an FD of 33%, with a 0.4 mm height. The scaffold coding is reported in Table 1. Compositions with the letter A correspond to scaffolds with an FD of 27% and larger spacing, while the letter B corresponds to an FD of 33% and smaller spacing.

Shape fidelity was considered a parameter to evaluate PCL/PEG blend printability. The shape fidelity of all polymeric blends was calculated using the following equation:(1)$$\mathrm{Shape}\;\mathrm{fidelity}=\frac{\mathrm{Mean}\;\mathrm{measured}\;\mathrm{diameter}}{\mathrm{Theoretical}\;\mathrm{diameter}}$$
where “measured diameter” refers to the diameter for the specific blend, measured with Fiji ImageJ 1.54f software (National Institute of Health, University of Wisconsin, Madison, WI, USA) [48] on images taken with a Leica Z16 APOA optical microscope equipped with a MC170 HD camera (Leica Microsystems Srl, Buccinasco (MI), Italy) and “theoretical diameter” refers to the nozzle diameter (400 µm).

To obtain the final PCL fibrous structure, 3D-printed PCL/PEG scaffolds were subjected to a porogen leaching process for 3 days in distilled water (DW, obtained with RiOs-DI^®^ 3 Water Purification System by Merck Life Science Srl (Milano, Italy), using 1 mL of DW per 10 mg of weighted sample, refreshed every day. The optimal leaching duration was calculated through a weight loss test (results shown in Appendix A). For each sample, the weight loss percentage was calculated at each time point using the following equation:(2)$$w_{l\%}=\frac{w_{in}-w_{n}}{w_{in}}\times 100$$
where *w_in_* is the weight of the sample before leaching, and *w_n_* is the weight at each leaching time point. Measurements were performed in triplicate.

50/50_A, 50/50_B, 40/60_A, and 40/60_B scaffolds revealed a fibrous morphology after the porogen leaching process (post-leaching), so they were defined as PCL fibrous scaffolds due to their morphology and the absence of PEG.

In order to achieve better biomimicry for cell culture experiments, PCL fibrous scaffolds were coated with porcine Gelatin (G) type A [38] and coded with the G suffix (G50/50_A, G50/50_B, G40/60_A, G40/60_B). Briefly, G was dissolved in a Phosphate-Buffered Saline (PBS) (Merk Life Science Srl) water solution while stirring at 40 °C for 1 h. The G solution was then cooled to room temperature before scaffolds were immersed in 3 mL of it overnight to allow protein deposition on the PCL surface.

### 2.5. PCL Fibrous Scaffold Charactrerizations

#### 2.5.1. Attenuated Total Reflectance–Fourier Transform Infrared (ATR-FTIR) Spectroscopy

ATR-FTIR spectroscopy was used to assess the complete removal of PEG from the 3D PEG/PCL scaffolds. A Nicolet™ iS50 FTIR spectrometer (Thermo Fisher Scientific, Monza, Italy) equipped with an ATR module and OMNIC™ Series software (Thermo Fisher Scientific) were used to analyze the samples. Neat materials (PCL and PEG) and PCL/PEG blend scaffolds pre- and post-leaching were analyzed in a spectral range of 525 and 4000 cm^−1^ with a resolution of 4 cm^−1^. Signals were collected in 32 scans, with a data spacing of 0.482.

#### 2.5.2. Scanning Electron Microscopy (SEM)

PCL fibrous scaffolds morphology was investigated through Scanning Electron Microscopy (SEM) using a beam of 10 keV at a 15 mm working distance using a TESCAN VEGA SEM platform (TESCAN Orsay Holdings, Brno, Czech Republic). All the samples were coated with gold for 2 min using an AGB7234 high-resolution sputter coating. Fiber diameters, fiber percentage distribution, and an analysis of post-leaching fiber alignment were obtained by analyzing SEM images using Fiji ImageJ software. Measurements were conducted, taking into account 50 fibers for each image.

#### 2.5.3. Mechanical Analysis

Tensile testing was conducted on PCL fibrous scaffolds using a QTest^®^/10 instrument (MTS^®^ System Corporation, Eden Prairie, MN, USA) and compared to PCL neat scaffolds, which were printed using only PCL with the same parameters as the PCL/PEG scaffolds. Tests were performed at a speed of 1 mm/min with a 50 N load cell, and data were analyzed using TestWorks^®^ 4 (version 4.12 F) software (MTS^®^ System Corporation). Stress–strain curves were plotted, and Young’s modulus was calculated as the slope of the line approximating their elastic part. Stress and strain were calculated using the following equations:(3)StressNmm2=LoadWidth×Thickness
(4)$$Strain\left[\%\right]=\frac{Elongation}{Initiallength}\times 100$$
where “width” (Equation (3)) refers to the infill mean value measured through ImageJ software on the PCL neat scaffolds, while for the post-leaching PCL fibrous scaffolds, it was considered 100% infill. For each group, 12 samples were tested.

#### 2.5.4. Atomic Force Microscopy Force (AFM) Spectroscopy Analysis

Mechanical characterization of the PCL fibrous scaffolds surfaces was performed by AFM force spectroscopy, also considering the presence of the G coating. A spherical indenter was assembled using a tipless cantilever (TL-CONT-20 by Nanosensors, Neuchatel, Switzerland) and a polystyrene sphere with about a 14.8 μm diameter (Flow Cytometry Size Calibration Kit F13838 by Thermo Fisher Scientific, bounded together using an epoxy adhesive cured by UV light. The cantilever spring constant was measured using thermal noise and the Sader-based method [49]. The estimation was repeated ten times, and the obtained cantilever spring constant was equal to 276.8 ± 5.3 mN/m considering its standard deviation, with a related resonance frequency equal to about 13.081 ± 0.007 kHz. The sensitivity was obtained by measuring the slope of the cantilever’s vertical deflection over the measured height of the piezo and pushing the tip over a glass microscopy slide taken as a non-deformable surface. This procedure was repeated ten times, and the sensitivity obtained was equal to 66.9 ± 0.35 nm/V, considering its standard deviation. The AFM piezo was set to obtain a 60 nN tip force over the sample during the extended segment experimental step. The extend and retract speeds were set to 50 μm/s, followed by a pause over the sample of about 200 ms. The elastic modulus was measured over a square grid of 100 points with a maximum measurable side length for the AFM of 100 μm, covering with a single measurement the widest possible area and obtaining more representativeness of the scaffolds’ surface elastic moduli. For each point, the measure was subsequently repeated 15 times. Measurements were performed in duplicate for each area, and five areas in different regions were selected for each scaffold type. Hertz’s model over retracted curves was applied to measure the local Young’s modulus of the scaffolds with the spherical punch model [50]. Measurements were performed in PBS at a room temperature of about 23 °C for all the scaffold types.

### 2.6. In Vitro Cell Culture

C2C12 immortalized myoblasts derived from murine satellite cells were used for an in vitro cell culture [39]. C2C12 cells were cultured in growth medium composed of Dulbecco’s Modified Eagle Medium (DMEM) (Thermo Fisher Scientific), 10% Fetal Bovin Serum (FBS) (Thermo Fisher Scientific), 1% penicillin/streptomycin (Thermo Fisher Scientific), and 2% L-glutamine (Thermo Fisher Scientific).

The G-coated and bare PCL fibrous scaffolds chosen for the in vitro cell culture (respectively, G40/60_A and 40/60_A) were sterilized under a biological hood in a 3-step process: (1) scaffolds were immersed in an ethanol (Carlo Erba Reagents) water solution (70% *v*/*v*) for 30 min, followed by PBS rinsing; (2) after drying, scaffolds were exposed to UV light for 30 min on each side; and (3) scaffolds were incubated overnight in a 2X antibiotic–antimycotic solution (Thermo Fisher Scientific) in PBS, followed by PBS rinsing. Direct C2C12 seeding on the surfaces of G-coated and bare PCL fibrous scaffolds was investigated. A total of 50,000 cells/cm^2^ were seeded (considering a seeding scaffold length of 20 mm) through the deposition of two drops of 25 µL on two different spots of each scaffold to allow for homogeneous seeding; after 1 h of incubation, the cell culture medium was added. After 7 and 14 days of culture, the cells were fixed with 4% *v*/*v* of Paraformaldehyde in PBS (Thermo Fisher Scientific) for 20 min and washed twice with PBS. The cells’ F-actin and nuclei were, respectively, stained with Phalloidin-AlexaFluor488 (Thermo Fisher Scientific) and 4′,6-diamidino-2-phenylindole (DAPI) (Thermo Fisher Scientific) diluted in 1% *w*/*v* of Bovine Serum Albumin (BSA, Merck Life Science Srl) in PBS. The cell culture experiment was first carried out over 7 days to observe the impact of the G coating on cell adhesion. Subsequently, to confirm the scaffolds’ capability to sustain myoblast fusion and differentiation, C2C12 cells were kept in the culture for 14 days on the G-coated scaffold G40/60_A. After fixation, the cells were immunostained with the primary antibody anti-myosin (#1570, Merck Life Science Srl) and the secondary antibody anti-mouse Alexa Fluor 555 (Thermo Fisher Scientific).

The samples were maintained in PBS until imaging with a Nikon Ti2-E fluorescence microscope (Nikon Instruments, Campi Bisenzio, Italy). Immunofluorescence experiments were performed in biological triplicate.

### 2.7. Statistic Analysis

Data are presented as mean values ± standard deviation. One-way ANOVA was performed with GraphPad Prism version 10.0.0 for Windows (GraphPad Prism 10 Software, Boston, MA, USA, www.graphpad.com), and differences were considered statistically significant when the *p*-values were lower than 0.05. Tukey’s post hoc test was conducted between two or more groups to determine if there were statistically significant differences between populations.

## 3. Results and Discussion

The objective of this study was to develop a 3D scaffold that mimics SMT morphology to support and stimulate the maturation of C2C12 cells without the need for additional external stimuli. The scaffold aligned structure was designed to provide topological cues crucial for replicating the anisotropy of SMT and inducing the formation of myotubes [18,22]. The proposed scaffold was fabricated by combining MEAM with porogen leaching, using PCL as the bulk material and PEG as the porogen [31,35]. To our knowledge, only one group has previously investigated the possibility of producing aligned PCL fibrous structures using PVA as the porogen to reproduce a muscle bundle [24,28]. Although the authors were able to obtain fibrous structures producing an anisotropic hierarchy of SMT and sustaining myotube formation, the manufacturing process involved several processing steps. In this work, we aimed at achieving the production of a PCL fibrous structure by employing a simple blending technique involving a one-step polymer blending process through a solvent casting technique at room temperature [37]. PCL/PEG blends were produced by solvent casting and used as the starting materials for the MEAM printing process. The PCL/PEG printed scaffolds were then subjected to porogen leaching in DW to produce PCL fibrous scaffolds. To evaluate the optimal process parameters (the polymer blend ratio and printing parameters), the post-leaching scaffolds morphologies were examined using optical imaging and SEM, while their mechanical properties were assessed through tensile testing and AFM. Finally, the optimized PCL fibrous scaffolds were coated with G to achieve a biomimetic coating for in vitro cell culture using C2C12 cells. The cells were cultured on the scaffold for 14 days to investigate the impact of the scaffolds 3D morphology, surface stiffness, and G coating on cell maturation and myotube formation. These effects were compared to a 2D control condition using a G-coated culture glass. Overall, the study aimed to demonstrate that the designed scaffolds could provide the necessary cues for the maturation of C2C12 cells into myotubes in a biomimetic 3D environment, potentially offering a more physiologically relevant approach compared to traditional 2D culture methods.

### 3.1. Polymeric Blend Characterization

PCL/PEG blend films were produced by solvent casting, evaluating three different polymeric weight ratios (PCL/PEG 60/40, 50/50, and 40/60).

DSC analyses were performed on samples of neat PCL and PEG films and on PCL/PEG films with the three different compositions (60/40, 50/50, and 40/60) to identify the suitable printing temperature for the MEAM process. A heating cycle (Figure 1c) from −90 °C to 150 °C was performed. All three PCL/PEG blends showed a single melting peak around 70 °C, comparable to the T_m_ of neat PCL (65 °C) and neat PEG (66 °C). Therefore, a printing temperature of 100 °C was selected to achieve homogeneous melting of the polymeric blends. Literature studies have also indicated that no degradation phenomena occur at this temperature [51,52]. The corresponding enthalpy values for PCL, PEG, and their blends are reported in Appendix A.

### 3.2. Three-Dimensional Scaffold Characterization

#### 3.2.1. Shape Fidelity of Pre-Leaching Scaffold

PCL/PEG blend films were used as the starting materials for the printing process. To investigate the effect of filament spacing on the leaching process and the formation of diverse densities of PCL fibers, the same blend compositions were printed using two different FDs (27% and 33%) to obtain a larger or smaller spacing between parallel filaments (Figure 1a).

All three blend compositions could be successfully printed, as shown in Figure 2 (upper row). The shape fidelity of the printing process was analyzed on images taken with an optical microscope (Figure 2, up) and calculated using Equation (1). Figure 1b compares the shape fidelity for each blend, showing values slightly higher than 1.5 for all the compositions and for both FDs. Increased fibers diameter, compared to the theoretical value (i.e., nozzle diameter = 400 µm), derived from the adopted printing parameters (pressure and PS) and were comparable with the results obtained with neat PCL (shape fidelity = 1.9). The average shape fidelity values of the blended scaffolds slightly decreased by increasing the PEG volumetric fraction (Figure 1b). This behavior may be attributed to the consistent printing parameters (such as pressure and PS) maintained across all compositions and porogen ratios without accounting for the differing viscosities of PCL and PEG due to their distinct molecular weights [53,54]. Moreover, within the same blend composition, no statistically significant differences were detected between the two FDs investigated.

#### 3.2.2. Leaching of 3D PCL/PEG Scaffolds

After printing, 3D PCL/PEG scaffolds were subjected to leaching process in DW for three days to completely remove the porogen (PEG). The leaching time was optimized by studying the weight variation during the leaching process (Appendix A).

To qualitatively evaluate the complete removal of PEG from the 3D scaffolds, ATR-FTIR analysis was conducted. Figure 3 shows the spectra of all blend scaffold compositions pre- and post-leaching compared with those of the scaffolds made of neat PCL and neat PEG. The spectral analysis of pre-leaching scaffolds revealed distinct adsorption bands indicative of the presence of PCL and PEG. The PCL characteristic adsorption bands are at 2943 and 2867 cm^−1^, corresponding to C-H symmetric stretching; at 1723 cm^−1^, representing C=O stretching; at 1292 cm^−1^, associated with C-O stretching; and at 1165 cm^−1^, indicative of C-O-C asymmetric stretching. Similarly, the presence of PEG is demonstrated by four characteristic adsorption bands: at 2879 cm^−1^, corresponding to C-H stretching; at 1145 cm^−1^, representing C-O stretching; at 1092 cm^−1^, indicating C-O-C stretching; and at 1059 cm^−1^, associated with C-O and C-C stretching [40,41]. On the contrary, the ATR-FTIR spectra of the post-leaching samples showed only PCL-related adsorption bands, confirming that PEG was completely removed during the leaching process.

#### 3.2.3. PCL Fibrous Scaffold Morphological Analysis

The morphological characterization of PCL fibrous scaffolds was carried out using optical microscopy and SEM.

Optical microscope images, shown in Figure 2, were used to evaluate macroscopic morphological changes in the scaffolds pre- and post-leaching. Fiber formation was correlated with the content of PEG in the starting blends. Leaching the scaffolds with the lowest ratio of PEG (60/40_A and 60/40_B) did not lead to the production of PCL fibers; therefore 60/40_A and 60/40_B samples were not considered for SEM morphological characterizations. 50/50 PCL/PEG scaffolds showed post-leaching fiber formation, but they were not homogeneously distributed. Finally, the scaffolds with the highest fraction of porogen (40/60_A and 40/60_B) produced the highest density of fibers, characterized by a highly homogeneous distribution. Moreover, for 50/50 and 40/60 compositions, the presence of a higher fill density (50/50_B and 40/60_B) led to the production of scaffolds with a higher fiber density. Fiber formation and distribution are highly dependent on the processes of phase separation and alignment that occur during extrusion. Previous studies have documented how factors such as extrusion pressure, temperature, polymer compatibility, and density disparities influence phase formation and alignment [12,31,35,55]. Under printing pressure, polymers with lower densities tend to migrate towards the outer regions, in contact with the syringe wall [55,56]. Additionally, in a polymeric blend, the component with the highest density or molecular weight will exert pressure on the other polymeric component, leading to the formation of a dispersed and not evenly distributed phase [31,35,57]. These phenomena could explain the absence of fiber formation in 60/40 scaffolds: the presence of PCL (with a higher molecular weight than PEG) as the major blend component hindered the formation of a uniform and aligned PEG phase within the printed filament, which was contrary to what was observed for 50/50 and 40/60 ratios (characterized by a higher content of PEG compared to PCL). 50/50 and 40/60 samples were characterized through SEM analyses, and the images are shown in Figure 4. The SEM analyses highlighted the increased homogeneity in fiber production for 40/60 samples compared to 50/50 samples, confirming the results from the optical images.

Moreover, SEM analyses confirmed that the combination of PS and pressure (2 mm/s and 10 kPa) was able to ensure an optimal post-leaching process resulting in the formation of fibrous PCL filaments without the presence of a PCL outer shell [42] (Appendix A). Optimization studies (Appendix A) indeed showed that printing at higher speeds and pressures induced the formation of a thin PCL shell around the struts, resulting in uneven and non-fibrous scaffolds. Such behavior was associated with the shear stresses inside the melted PCL/PEG polymeric blend during the printing process, which probably caused the formation of the outer PCL shell [42].

The percentage distribution of fiber diameters (Figure 5a–d) was calculated starting from SEM images at a higher magnification (not reported) using Fiji ImageJ software. 40/60 scaffolds showed thinner and more homogeneous fibers, with a prevalence of fibers between 1 µm and 10 µm in diameter. On the other hand, 50/50 scaffolds showed a non-homogeneous fiber diameter distribution, with a relevant presence of fibers with a diameter over 60 µm, some of them reaching hundreds of µm (up to 600 µm). Moreover, 50/50_B and 40/60_B scaffolds showed a higher percentage of fibers with the smallest diameters of 1–10 µm (9% for 50/50_B scaffold compared to 0% for 50/50_A scaffold; 63% for 40/60_B scaffold compared to 47% for 40/60_A scaffold). These results confirmed the qualitative evaluation obtained through optical images.

Finally, fiber alignment and directionality were evaluated from SEM images through distribution analyses, as shown in Figure 5e–h. All the scaffolds resulted in an even distribution of fiber directionality, aligned along the printing direction (around 100°). The scaffolds with the lower FD (50/50_A and 40/60_A) showed a higher alignment than those with the higher FD (50/50_B and 40/60_B).

Considering Figure 5a–d, 40/60_A scaffold architecture, characterized by fiber dimeters prevalently between 1 and 20 μm, was most similar to both small muscle fibers (20 μm) [6] and myofibril (1–2 μm) [5]; thus, it was better able to mimic SMT microscale morphology in a 3D environment.

Overall, with the proposed scaffold fabrication strategy, we achieved aligned fiber formation with a relatively low concentration of porogen, compared to the literature, and with small fiber diameters. Size distribution and homogeneity depend on filament spacing (varying with FD), but mostly on the porogen (PEG) volume ratio. Previous work based on PVA and PCL demonstrated the dependence between the porogen amount and fiber formation: PVA/PCL blends with higher porogen (PVA, Mw = 89–98 kDa) concentrations (e.g., 70:30) hindered aligned pattern formation due to viscosity differences between PCL (Mw = 45 kDa) and PVA. On the other hand, a lower porogen–polymer proportion allowed to achieve aligned fiber formation [35]. In our case, however, the porogen (PEG, Mw = 35 kDa) has a lower molecular weight than PCL (Mw = 43 kDa); hence, we observed an opposite behavior, in which PCL/PEG blends with a lower porogen ratio (<50%) did not allow fiber formation.

#### 3.2.4. Mechanical Analyses of PCL Fibrous Scaffolds

Mechanical properties were evaluated through tensile testing conducted on PCL neat scaffolds (100/0_A and 100/0_B), used as a control, and all the PCL fibrous scaffolds. The stress/strain curves of the scaffolds are reported in Figure 6a. In general, the PCL scaffolds, after the porogen leaching process, showed poor mechanical properties compared to the neat PCL ones, sustaining shorter deformation (around 5% compared to the 9–10% strain for PCL) and lower stress (3.5–4.5 MPa compared to 6–7 MPa). This behavior, more relevant in fibrous scaffolds, is a result of porogen phase removal after the leaching process and is comparable to literature results [28].

The elastic modulus (Young’s modulus) was calculated as the slope of the line approximating the elastic part of the stress/strain curve. Overall, all the blends evaluated displayed a decrease in the Young’s modulus compared to the neat PCL scaffolds (from around 350 kPa to around 150 kPa) (Figure 6c) due to the removal of the porogen volume fraction and the formation of the aligned fibrous structure, as demonstrated by the previously discussed morphological analysis. Additionally, no significant differences were observed using different FDs. Between the different blends, the 60/40 composition showed the highest Young’s modulus, due to the absence of the formation of smaller fibers (Figure 2). No statistically significant differences were detected between 50/50 and 40/60 samples. Figure 6b compares the stress/strain curves in the elastic interval (up to 1% deformation) for all the blends. All the PCL fibrous scaffolds showed a so-called “J-shape” in the stress/strain curve, probably due to a slight deformation remaining on some of the fibers after the leaching process [43].

Although tensile testing offers insights into the overall mechanical properties of a scaffold as a cohesive structure, when examining scaffold suitability as an in vitro model under static conditions, surface stiffness represents the critical factor influencing cell behavior [13]. For this reason, PCL fibrous scaffolds (50/50 and 40/60 post-leaching) were further characterized using Atomic Force Microscopy (AFM).

AFM was performed on both bare PCL fibrous scaffolds and G-coated scaffolds (coded with G) (i.e., biomimetic coating to improve PCL fibrous scaffolds’ bioactivity for in vitro cell studies), aiming to evaluate the mechanical properties of scaffolds surfaces and the influence of the biomimetic coating. Both bare and G-coated PCL fibrous scaffolds showed a statistical distribution of the Young’s modulus ranging from 10 to 80 kPa (Figure 7). These values follow within the range of SMT stiffness properties (10–60 kPa [15,16]) obtained in the literature using the AFM technique. In particular, G40/60_A and G40/60_B scaffolds demonstrated values between 10 and 15 kPa, which are comparable to the surface stiffness identified in the literature as optimal to promote myotube formation [13]. Finally, Young’s modulus values for all the types of PCL fibrous scaffolds were not influenced by differences in fiber size and density that were observed through the morphological analysis among the different porogen ratios (Figure 2, Figure 4 and Figure 5).

### 3.3. In Vitro Cell Culture

40/60_A scaffolds were selected for in vitro cell culture validation as they showed a denser and more homogeneous distribution of fiber diameters after the morphological evaluation. The scaffolds were coated with G to achieve a biomimetic surface composition as well as proper surface stiffness, as evaluated by the AFM analysis. G40/60_A scaffolds were seeded and cultured with C2C12 cells. In parallel, uncoated 40/60_A scaffolds were used as controls to highlight the need for G coating, while C2C12 cells seeded on cell culture-treated polystyrene (Jet BioFil) with the same concentration (50,000 cells/cm^2^) were used as 2D G-coated controls to underline the scaffolds anisotropic effects on cells orientation and myotube formation. Cell culture studies were conducted for 7 days in order to clearly assess the effect of the G coating on cell adhesion and for 14 days to evaluate the effect of PCL fibrous scaffolds morphology on cell arrangement and myotube formation. Figure 8 shows confocal images of the cells stained for F-actin and nuclei after a 7-day culture. The G coating led to enhanced cell adhesion and proliferation on the scaffolds micrometric fibers (Figure 8c) compared to the uncoated condition (Figure 8b), where a few cells were scatteredly attached. Moreover, comparing Figure 8c with the 2D control on polystyrene (Figure 8a), cells were arranged in an aligned morphology, following scaffold filaments orientation after 7 days, while no cell alignment was detected in the 2D control. This evidence demonstrates that the PCL fibrous scaffolds, thanks to their morphology and aligned fibers, can drive and maintain cells orientation for up to 7 days when coupled with a simple protein coating. Since the G coating ensured proper cell adhesion, C2C12 cells were seeded onto the G40/60_A scaffolds for 14 days to evaluate the capability of fibrous aligned morphology to influence C2C12 differentiation through an investigation of myotube formation using immunostaining against myosin (Figure 9), a protein expressed by differentiated muscle cells. After 14 days, aligned and fully formed myotubes were observed along the G40/60_A scaffolds’ fibers (Figure 9b). On the other hand, cells in the 2D culture control formed shorter and not fully aligned myotubes (Figure 9a). This result highlighted the capability of 3D topographical stimuli to influence myotube formation. Previous studies have achieved cell alignment on 3D PCL microfibers through collagen coating [28]. The use of collagen coatings or hydrogels, although highly biomimetic, involves high costs [36]. In this work, biomimetic coating was achieved through the deposition of G, which, as a collagen derivative, provides similar stimuli [14] to cells but has a lower cost [35].

## 4. Conclusions

In this study, 3D hierarchical constructs based on PCL with unidirectionally aligned microfibers were successfully engineered using a combination of MEAM and porogen leaching. PCL was blended with PEG, used as porogen, in varying ratios (60/40, 50/50, 40/60% *w*/*w*) through a solvent casting technique. Thermal characterization was conducted to assess the suitability of the PCL/PEG blends as materials for MEAM processes. The scaffold design comprised 30 mm long parallel filaments with variable spacing, ranging from 500 to 550 µm. The printing parameters were optimized to induce polymeric phase alignment, avoiding the formation of a polymeric PCL shell around the printed filaments. Leaching in bi-distilled water for 3 days effectively removed PEG. Morphological analyses of the post-leaching scaffolds indicated the absence of fiber formation in the 60/40 scaffolds, while the 50/50 and 40/60 scaffolds exhibited well-aligned fibers, with the 40/60 scaffold displaying a more uniform distribution and smaller diameters (ranging from 10 to 60 µm). The mechanical characterization through tensile testing revealed lower Young’s moduli in the post-leaching scaffolds compared to neat PCL. The scaffolds showing fiber formation, called PCL fibrous scaffolds, were coated with G to achieve a more biomimetic surface composition. AFM analyses showed that all types of fibrous scaffolds coated with G have stiffness within the range of SMT values (measured with the same technique). Considering the morphological and mechanical results, 40/60_A scaffold composition was chosen for cell culture validation tests as it showed an optimal fiber distribution and suitable stiffness values (10–20 kPa). Before biological evaluation, the 40/60_A scaffold was coated with G (G40/60_A) to increase the PCL bioactivity. In vitro cell culture experiments using C2C12 murine myoblasts demonstrated the efficacy of the G coating to favor cell attachment compared to bare PCL. Moreover, cell culture results highlighted the capacity of a microfiber morphology to induce cellular alignment and myotube formation. Future experiments will involve a longer cell culture duration to allow for the better maturation of C2C12 cells and myotube formation. Moreover, more accurate biological tests will be conducted to assess proper cell maturation and completely validate the SMT in vitro model. This platform holds potential for in vitro modeling of various human tissues with anisotropic morphology. For instance, in addition to SMT, aligned fibrous PCL scaffolds could be employed to simulate the structure and mechanical behavior of aligned tissues such as tendons, ligaments, and nerve fibers [48,58,59]. By mimicking the specific mechanical properties and surface stiffness of these tissues, these scaffolds will prove useful not only for in vitro modeling but also for applications in tissue engineering and regenerative medicine.

## Figures and Tables

**Figure 1 bioengineering-11-00332-f001:**
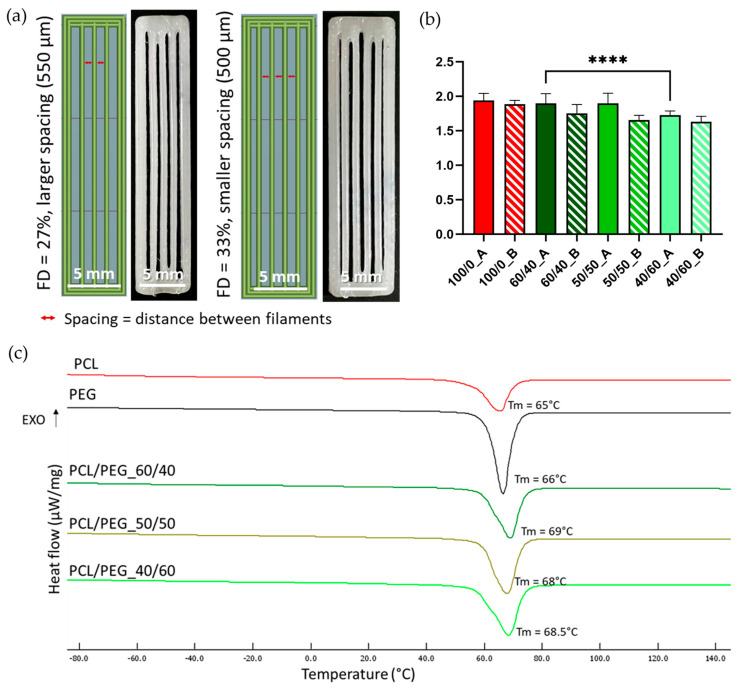
(**a**) CAD images (left) and photos (right) of scaffolds with aligned filaments, designed and printed with different FDs and corresponding to different filament spacings. (**b**) Shape fidelity values for all scaffolds printed with PCL/PEG blends and neat PCL. Nozzle diameter was taken as the theoretic value reference. N = 3 (**** *p* < 0.0001). (**c**) DSC thermographs of the heating cycle (with indication of the corresponding values of Tm) of neat PCL (red line), neat PEG (black line), and PCL/PEG, 60/40 (dark green line), 50/50 (maroon line), and 40/60 (light green line) blends.

**Figure 2 bioengineering-11-00332-f002:**
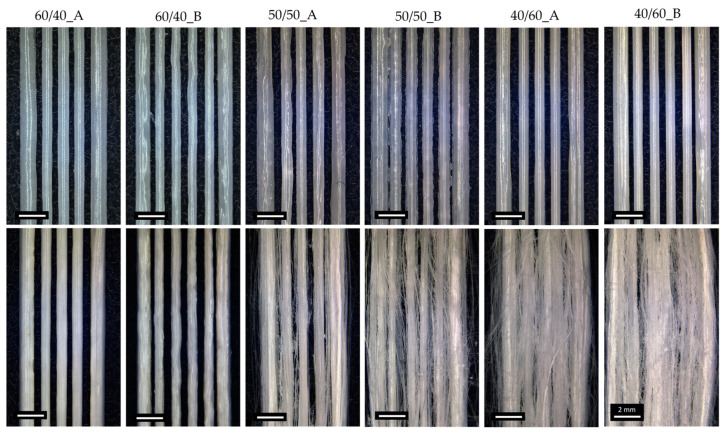
Optical images of 3D-printed PCL/PEG scaffolds before (**up**) and after (**down**) the leaching process, showing the effect of the blend composition and FD on fiber formation.

**Figure 3 bioengineering-11-00332-f003:**
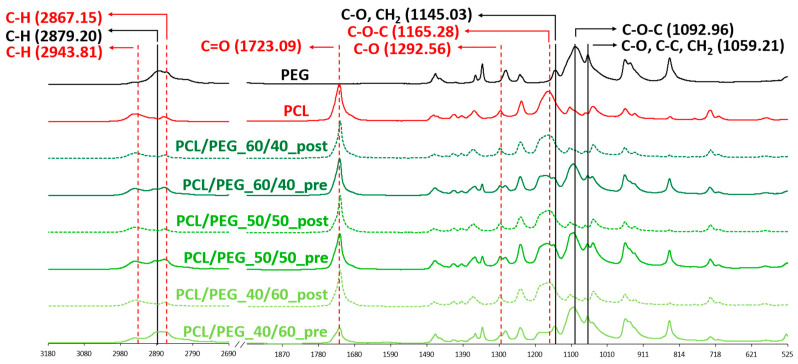
ATR-FTIR spectra of neat PEG and PCL compared to PCL/PEG 60/40, PCL/PEG 50/50, and PCL/PEG 40/60 scaffolds before (pre) and after (post) leaching. Absorption peaks associated with PCL are colored in red; absorption peaks associated with PEG are labelled in black.

**Figure 4 bioengineering-11-00332-f004:**
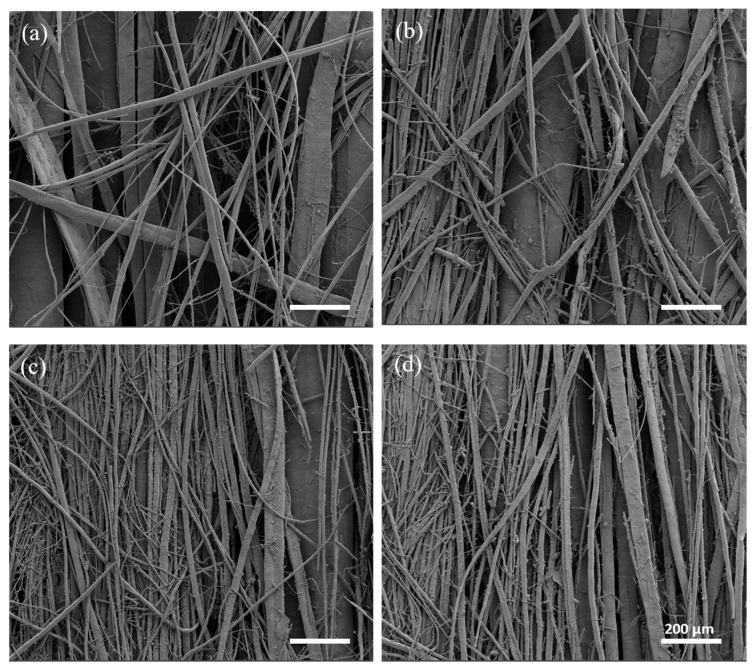
SEM images taken at a high magnification (100×): (**a**) 50/50_A, (**b**) 50/50_B, (**c**) 40/60_A, and (**d**) 40/60_B.

**Figure 5 bioengineering-11-00332-f005:**
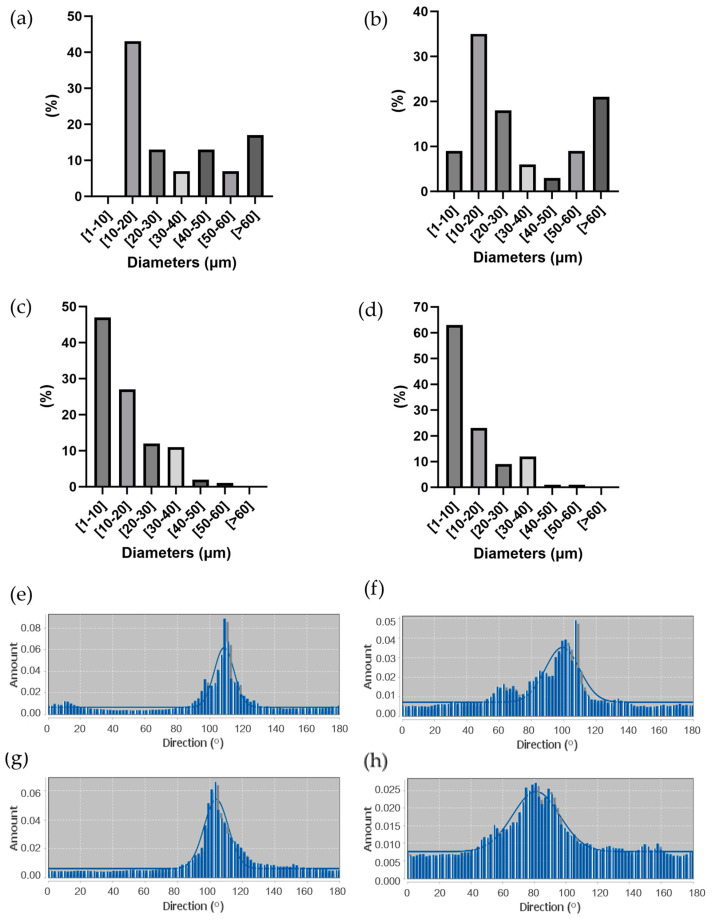
(**a**–**d**) Percentage distribution of fiber diameters measured through ImageJ analysis of SEM images of (**a**) 50/50_A, (**b**) 50/50_B, (**c**) 40/60_A, and (**d**) 40/60_B. (**e**–**h**) Directional distribution of fibers in PCL fibrous scaffolds along printing direction (100 °C) for (**e**) 50/50_A, (**f**) 50/50_B, (**g**) 40/60_A, and (**h**) 40/60_B.

**Figure 6 bioengineering-11-00332-f006:**
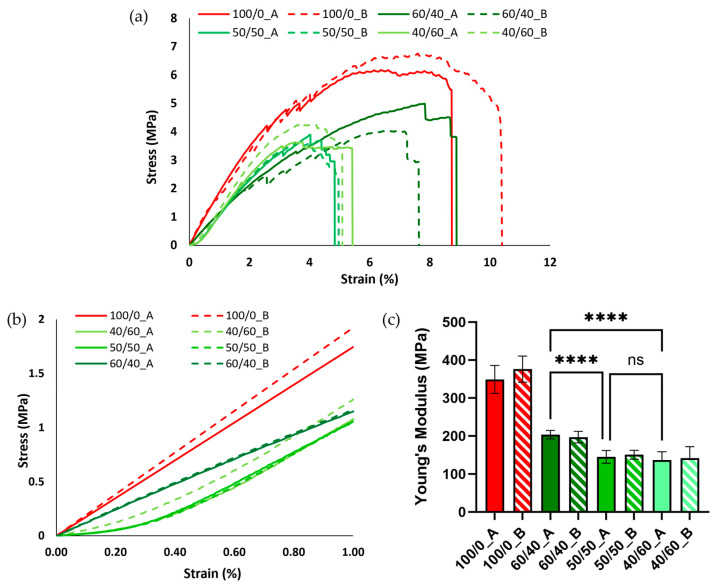
(**a**) Stress/strain curves in the elastic trait (1% strain) for neat PCL and PCL fibrous scaffolds. (**b**) Stress/strain curves in the elastic trait (1% strain) for neat PCL and PCL fibrous scaffolds. (**c**) Young’s modulus comparison between neat PCL and PCL fibrous scaffolds (**** *p* < 0.0001; ns = not statistically significant).

**Figure 7 bioengineering-11-00332-f007:**
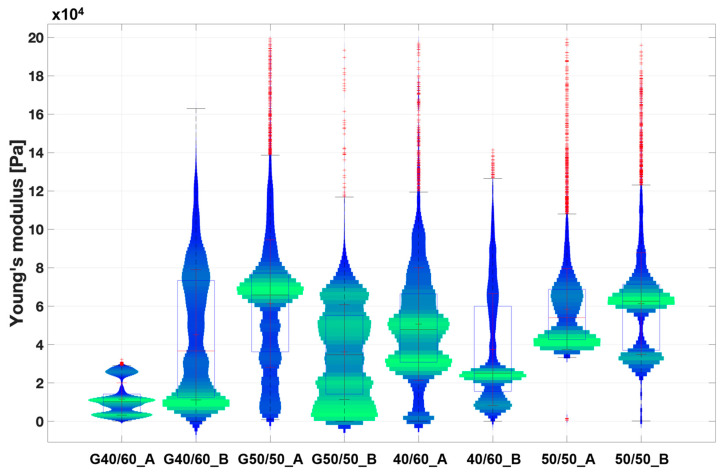
Violin plot of the extracted Young’s modulus from the collected extended curves of PCL fibrous scaffold samples in wet conditions. Scaffolds coated with Gelatin are coded with G. The graph illustrates the distribution of the measured Young’s modulus values across various scaffold surface areas. A broader and more pronounced green range indicates a higher frequency of measured values within that range, while blue range represents less frequently detected values Finally, each red cross stands for an outlier value.

**Figure 8 bioengineering-11-00332-f008:**
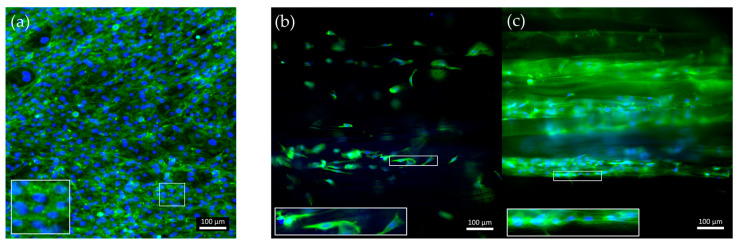
Confocal images of F-actin staining after a 7-day culture of C2C12 cells on (**a**) control-treated polystyrene, (**b**) uncoated, and (**c**) G-coated G40/60_A PCL fibrous scaffolds. Green: F-actin; blue: nuclei.

**Figure 9 bioengineering-11-00332-f009:**
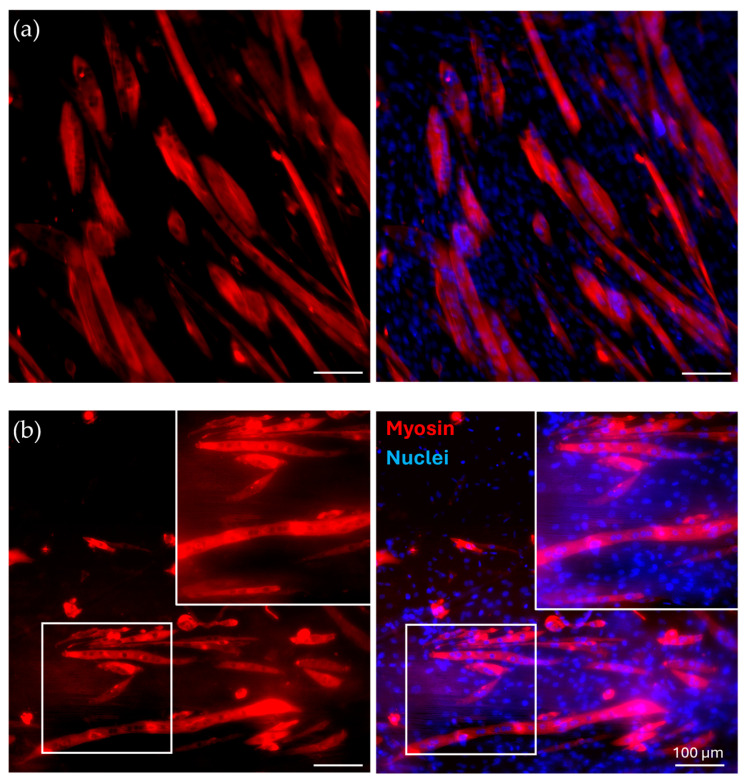
Immunofluorescence for myosin. C2C12 cells were cultured for 14 days in (**a**) control conditions and on (**b**) G-coated G40/60_A PCL fibrous scaffolds, with zoomed-in, detailed pictures. Red: myosin; blue: nuclei.

**Table 1 bioengineering-11-00332-t001:** Coding for different scaffolds produced. Letter A corresponds to scaffolds with larger filament spacing (550 µm, FD 27%), while letter B corresponds to smaller filament spacing (500 µm, FD 33%).

Code ^1^	PCL/PEG Ratio (*w*/*w*)
100/0_A	100/0
100/0_B	100/0
60/40_A	60/40
60/40_B	60/40
50/50_A	50/50
50/50_B	50/50
40/60_A	40/60
40/60_B	40/60

^1^ corresponds to the different densities of PCL fibers formed after the leaching process.

## Data Availability

Data will be available on request.

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
