# Peer review of "Three-Dimensional Microfibrous Scaffold with Aligned Topography Produced via a Combination of Melt-Extrusion Additive Manufacturing and Porogen Leaching for In Vitro Skeletal Muscle Modeling"

_bioengineering, 2024, doi:10.3390/bioengineering11040332_

Round 1

Reviewer 1 Report

Comments and Suggestions for Authors

The presented paper is of high quality. The authors presented a very interesting concept. All the experiments are well-planned. The discussion is quite exhaustive. I recommend this manuscript for publication without changes. 

Author Response

We thank the reviewer for her/his work and for appreciating the manuscript.

Reviewer 2 Report

Comments and Suggestions for Authors

The manuscript describes 3D printing of PCL/PEG where PEG is a sacrificial component to generate fibrillar architecture similar to native muscle tissue. While the concept is not highly innovative the formed constructs are interesting. However, there are important concerns associated with the study that should be addressed:

- The selection of PCL for muscle tissue engineering is not justified as the mechanical properties and biological activity are not suitable. Limitations of the study should be highlighted.

- The mechanical properties of the scaffolds should be compared to native tissue.

- The biological experiments are too basic. Scaffolds are not optimized based on the biological outcome, the growth rate and differentiation of myoblasts are not explored. Without performing cellular studies the title is misleading. 

Comments on the Quality of English Language

Overall the text is easy to follow, a quick proofreading can further improve the text.

Reviewer 3 Report

Comments and Suggestions for Authors

1.       PCL scaffolds with aligned and cross-linked structure have been obtained by electrospinning technique. It is one of the most frequently-used methods which enable fast and controlled fiber alignment, e.g. reported extensively elsewhere DOI 10.1088/1757-899X/98/1/012024, thus I would recommend the authors providing some more details on the advantages of aligned fibers for tissue engineering applications.

2.       In general, aligned scaffolds reveal decreased mechanical properties compared with cross-linked scaffolds. This point should be addressed in the manuscript in more detail.

3.       I would recommend providing results as average and standard deviations. Otherwise the significance of the obtained differences is not obvious.

4.       In Fig. 5a it is seen that there are no fibers with diameter lower than 10 mkm, however, I believe that such fibers are present if other magnification images are used. If not, just provide a brief comment.

5.       I would also recommend providing some brief analysis on the mechanical compatibility of the scaffolds with the targeted tissue, these materials are intended to replace.

6.       I would recommend to concentrate discussion on the results obtained rather than repeating experiments done in the manuscript.

7.       The intensities of the melting peaks are always different, provide some details on the possible reasons and mechanisms.

Reviewer 4 Report

Comments and Suggestions for Authors

The authors presented a combination of melt extrusion and salt leaching for the fabrication of fibrous scaffolds for muscle tissue engineering. The authors deeply characterized the obtained scaffolds, testing multiple parameters combination. However, few biological data are presetned.

Major:

·         Figure 2: the photos of the scaffold do not represent the printing preview of Figure 1a. Please include an entire photo, do not cut it.

·        Please reorganize the figure. The photo of the scaffolds (Figure 2a) should be in the same panel of the printing preview (Figure 1a) and of the shape fidelity (Figure 1b). Moreover, the order of the sub-figures (A,B,C) must be as they appear in the text.

·       The authors should describe more in details the physical process that leads to the formation of the microfibers, comparing with what was reported in literature. Moreover, additional comments should be provided on the reason why on some samples the behavior was not observed.

·       The authors should include additional biological validation. They should provide proliferation data on multiple time points, to show the ability of cells to grow/live on the scaffold. Moreover, since the authors aims at exploiting the scaffold as an in vitro models of muscle for drug testing, they should add a proof-of-concept for it (e.g., show the effect knowm molecules) 

Minor:

·         Table 1: Please include the difference between the A and B, since it not clear

Round 2

Reviewer 3 Report

Comments and Suggestions for Authors

can be accepted now.

Reviewer 4 Report

Comments and Suggestions for Authors

The authors thoroughly answered to all my comments